# Isolation and Characterization of Colistin-Resistant *Enterobacteriaceae* from Foods in Two Italian Regions in the South of Italy

**DOI:** 10.3390/microorganisms13010163

**Published:** 2025-01-14

**Authors:** Rosa Fraccalvieri, Angelica Bianco, Laura Maria Difato, Loredana Capozzi, Laura Del Sambro, Stefano Castellana, Adelia Donatiello, Luigina Serrecchia, Lorenzo Pace, Donatella Farina, Domenico Galante, Marta Caruso, Maria Tempesta, Antonio Parisi

**Affiliations:** 1Experimental Zooprophylactic Institute of Apulia and Basilicata, 71121 Foggia, Italy; rosa.fraccalvieri@izspb.it (R.F.); maria.difato@izspb.it (L.M.D.); loredana.capozzi@izspb.it (L.C.); laura.delsambro@izspb.it (L.D.S.); stefano.castellana@izspb.it (S.C.); adelia.donatiello@izspb.it (A.D.); luigina.serrecchia@izspb.it (L.S.); lorenzo.pace@izspb.it (L.P.); donatella.farina@izspb.it (D.F.); domenico.galante@izspb.it (D.G.); marta.caruso@izspb.it (M.C.); antonio.parisi@izspb.it (A.P.); 2Department of Veterinary Medicine, University Aldo Moro of Bari, Strada per Casamassima Km 3, 70010 Bari, Italy; maria.tempesta@uniba.it

**Keywords:** antimicrobial resistance (AMR), colistin, *Enterobacteriaceae*, mcr gene, multidrug resistance (MDR)

## Abstract

The emergence of colistin-resistant *Enterobacteriaceae* in food products is a growing concern due to the potential transfer of resistance to human pathogens. This study aimed to assess the prevalence of colistin-resistant *Enterobacteriaceae* in raw and ready-to-eat food samples collected from two regions of Italy (Apulia and Basilicata) and to evaluate their resistance phenotypes and genetic characteristics. A total of 1000 food samples were screened, with a prevalence of 4.4% of colistin-resistant *Enterobacteriaceae*. The majority of the isolates belonged to *Enterobacter* spp. (60%), followed by *Moellerella wisconsensis*, *Atlantibacter hermannii*, *Klebsiella pneumoniae*, and *Escherichia coli*, among others. Genomic sequencing and antimicrobial susceptibility testing revealed high levels of resistance to β-lactams, with most isolates exhibiting multidrug resistance (MDR). Notably, seven isolates harbored *mcr* genes (*mcr*-1, *mcr*-9, and *mcr*-10). Additionally, in four of them were predicted the IncHI2 plasmids, known to facilitate the spread of colistin resistance. Furthermore, 56 antimicrobial resistance genes were identified, suggesting the genetic mechanisms underlying resistance to several antibiotic classes. Virulence gene analysis showed that *E. coli* and other isolates carried genes linked to pathogenicity, increasing the potential risk to public health. This study emphasizes the role of food as a potential reservoir for colistin-resistant bacteria and the importance of monitoring the spread of AMR genes in foodborne pathogens.

## 1. Introduction

Antimicrobial resistance bacteria represent one of the most important challenges for public health worldwide. The *Enterobacteriaceae* family are Gram-negative bacteria, including the genera *Escherichia*, *Klebsiella*, *Salmonella*, *Shigella*, *Citrobacter*, and *Enterobacter* which are often associated with nosocomial and community-acquired infections, such as pneumonia, bacteremia, and respiratory or urinary tract or abdominal infections, among others [1,2,3,4,5]. The incidence of multidrug resistance (MDR) *Enterobacteriaceae*, including strains producing extended-spectrum β-lactamases (ESBLs) and those resistant to colistin and carbapenems, is increasing [6,7,8] and is frequently found in food and the environment [9,10,11].

Colistin, used to treat intestinal infections in livestock, is a last resort antibiotic for the treatment of human infections caused by multidrug-resistant Gram-negative bacteria, particularly carbapenem-resistant strains in humans [12]. Colistin is categorized in the World Health Organization (WHO) List of Critically Important Antimicrobials (CIA) as a highest priority critically important antimicrobial (HstPCIA) for human medicine (WHO CIA List) [13]. Resistance to colistin is associated with both chromosomal mutations and the acquisition of mobile colistin resistance (*mcr*) genes carried by plasmids [12,14]. In 2015, Liu et al. discovered the plasmid-mediated *mcr*-1 gene, associated with colistin resistance in *Escherichia coli* [15]. Since then, other *mcr* variants, including *mcr*-2, *mcr*-3, *mcr*-4, *mcr*-5, *mcr*-6, *mcr*-7, *mcr*-8, *mcr*-9, and *mcr*-10, have been identified in various *Enterobacteriaceae* species [16,17,18,19,20,21,22,23,24]. These *mcr* genes have been detected in *E. coli*, *Enterobacter cloacae*, *Klebsiella pneumoniae*, *Salmonella enterica*, *Citrobacter freundii*, *Citrobacter braakii*, and *Raoultella ornithinolytica* strains isolated from food, the environment, animals, and humans [25]. The dissemination of these strains in food represents a potential hazard to human health, since the genes encoding these resistances are often associated with motile genetic elements that can be transferred to both commensal and pathogenic bacteria, including those of other genera and species [26]. The dominant plasmids contributing to the worldwide spread of the *mcr* gene in *Enterobacteriaceae* include IncI2, IncX4, IncP, and IncHI2 [19,27].

In order to increase the knowledge about the prevalence of colistin-resistant *Enterobacteriaceae* in food, this study aims to screen 1000 food samples collected from two regions, Apulia and Basilicata. Therefore, we will assess the antibiotic resistance of the isolated strains using a traditional approach. In addition, whole genome sequencing (WGS) will allow us to predict antibiotic resistance genes. The comparison of the results obtained through the two different approaches will help to classify organisms in terms of colistin resistance and increase the knowledge on this topic.

## 2. Materials and Methods

During 2018 and 2023, 1000 several food samples (500 raw foods; 500 ready-to-eat (RTE)) were collected and analyzed. The raw food samples included milk (*n* = 52), meat (*n* = 207), seafood products (*n* = 133), bakery and pastry products (*n* = 28), and vegetables (*n* = 80). The RTE samples included milk and dairy products (*n* = 240), dried or cooked sausages (*n* = 35), ready meals (i.e., gastronomic preparations based on meat or fish or vegetables such as grilled swordfish) (*n* = 100), bakery and pastry products (i.e., cream puffs and sweets) (*n* = 27), ice creams (*n* = 66), and vegetables (*n* = 32).

### 2.1. Isolation of Bacterial Strains

The food samples were analyzed to detect colistin resistance *Enterobacteriaceae*, as follows: 5 g or 5 mL of each sample was added to 45 mL of buffered peptone water (BPW, Biolife, Milano, Italy), homogenized for 30 s using a Stomacher (Seward, Worthing, United Kingdom), and incubated at 37 °C for 18 ± 2 h. After incubation, 1 mL of the broth was inoculated into 10 mL of EE Broth Mossel 1X (Liofilchem^®^, Italy) and incubated at 37 °C for 24 ± 2 h. Then, 10 μL of the broths were plated onto chromogenic agar CHROMagar™ COL-APSE (CHROMagar, Paris, France) and incubated at 37 °C for 24 ± 2 h. Colistin-resistant colonies developed species-specific colors (reddish to dark pink coloration for *E. coli* and metallic blue coloration for *Citrobacter* spp., *Klebsiella* spp., and *Enterobacter* spp.).

### 2.2. Colistin-Resistant Screening

All typical presumptive colistin resistance *Enterobacteriaceae* strains previously isolated were subcultured with the aim to detected colistin resistance (COL, 0.0625–64 µg/mL) by the antimicrobial minimum inhibitory concentration (MIC), using the “MIC Strip Colistin” kit (MERLIN-A Bruker Company, Singapore), according to the Clinical and Laboratory Standards Institute (CLSI) [28].

### 2.3. Whole Genome Sequencing (WGS)

Entire DNA was purified using the QIAmp DNA mini kit (Qiagen, Hilden, Germany), according to the manufacturer’s instructions. The concentration of purified DNA was evaluated by Qubit Fluorometers 3.0 using the Qubit dsDNA HS Assay (Thermo Fisher Scientific). 

Libraries were prepared using the Illumina DNA Prep kit and sequencing using a MiSeq Reagent kit V2 (500 cycles) (Illumina, San Diego, CA, USA) on a MiSeq platform, as reported in Bianco et al. 2021 [29]. Bioinformatic analyses were performed using the European Galaxy server (https://usegalaxy.eu/ accessed on 15 July 2024)). The raw data were assembled by SPAdes v3.15 [30], and the assembly quality check was analyzed by Quast v.5.0.2 [31]. Ribosomal MLST (rMLST; https://pubmlst.org/species-id; accessed on 19 July 2024)) was employed for typing the isolates. AMR genes, virulence genes, and plasmids were predicted using the ABRIcate Mass screening tool (Galaxy Version 1.0.1); default parameters were imposed on the abovementioned tools. 

### 2.4. Antimicrobial Susceptibility Test

Strains identified as colistin-resistant *Enterobacteriaceae* were tested by the minimum inhibitory concentration (MIC) method according to the Clinical and Laboratory Standards Institute [28] for Enterobacterales, using the Sensititre™ panels GN4F^®^, CMV4AGNF^®^, and EUVSEC^®^ (Trek Diagnostic Systems, Westlake, OH, USA). The EUVSEC^®^ panel includes the combination test method to detect ESBL producers [28]. The antimicrobials tested are listed below: ampicillin (AMP 1–32 µg/mL), piperacillin (PIP 16–64 µg/mL), amoxicillin/clavulanic acid 2:1 ratio (AUG2 1/0.5–32/16 µg/mL), ampicillin/sulbactam (A/S2 4/2–16/8 µg/mL), piperacillin/tazobactam constant 4 (P/T4 8/4–128/4 µg/mL), ticarcillin/clavulanic acid constant 2 (TIM2 8/2–64/2 µg/mL), azithromycin (AZI 0.25–32 µg/mL), amikacin (AMI 0.25–32 µg/mL), gentamicin (GEN 0.25–16 µg/mL), streptomycin (STR 2–64 µg/mL), tobramycin (TOB 2–8 µg/mL), chloramphenicol (CHL 2–32 µg/mL), ciprofloxacin (CIP 0.015–4 µg/mL), levofloxacin (LEVO 1–8 µg/mL), nalidixic acid (NAL 0.5–32 µg/mL), cefazolin (FAZ 1–16 µg/mL), cefoxitin (FOX 0.5–64 µg/mL), ceftriaxone (AXO 0.25–64 µg/mL), cefepime (FEP 0.06–32 µg/mL), minocycline (MIN 1–8 µg/mL), tetracycline (TET 4–32 µg/mL), tigecycline (TGC 1–8 µg/mL), sulfisoxazole (FIS 16–256 µg/mL), trimethoprim/sulfamethoxazole (SXT 0.12/2.38–4/76 µg/mL), aztreonam (AZT 1–16 µg/mL), meropenem (MERO 0.03–16 µg/mL), doripenem (DOR 4–8 µg/mL), ertapenem (ETP 0.25–8 µg/mL), imipenem (IMI 0.12–16 µg/mL), cefotaxime (FOT 0.25–64 µg/mL), and ceftazidime (TAZ 0.25–128 µg/mL). Cefotaxime/clavulanic acid (F/C 0.06/4–64/4 µg/mL) and ceftazidime/clavulanic acid (T/C 0.12/4–128/4 µg/mL) were used for the combination test method.

*E. coli* ATCC 25922 was used as the control strain. The MIC values were interpreted using the guidelines described in the Clinical and Laboratory Standards Institute [28].

Bacteria with resistance to at least one antimicrobial in at least three classes of antibiotics were defined as multidrug-resistant (MDR) [32]. 

### 2.5. Nucleotide Sequence Accession Numbers

The genomes sequenced were available in GenBank as BioProject PRJNA1076372 (Appendix A).

### 2.6. Statistical Analysis

The chi-square test (v2, *p* < 0.05) with Epi Info 3.3.2 software was used to compare the differences among the percentages of several isolates.

## 3. Results

### 3.1. Isolation of Colistin Resistance Bacteria

In total, 1000 food samples were tested and colistin resistance *Enterobacteriaceae* strains were cultured from 5% (25/500) of the raw food samples and 4% (19/500) of the ready-to-eat food samples. No statistically significant difference was observed between the two groups (*p* > 0.05).

A total of 45 colistin-resistant *Enterobacteriaceae* strains were isolated from 44 (4.4%; 44/1000) positive food samples. Specifically, 26 strains were cultured from 25 raw food sample; two isolates were detected from the same source (Table 1 and Table 2).

Genomic sequencing of the strains isolated and rMLST analysis of the draft genome allowed for species identification (Table 1 and Table 2). The most frequently isolated species was *Enterobacter* spp. (27/45; 60%), followed by *Moellerella wisconsensis* (7/45; 16%), *Atlantibacter hermannii* (4/45; 9%), and *Klebsiella* spp. (2/45; 4%). Strains of *Escherichia coli*, *Citrobacter freundii*, *Raoultella ornithinolytica*, *Kosakonia cowanii*, and *Phytobacter diazotrophicus* were also isolated, with low frequencies.

The isolation source of the colistin-resistant strains, their species, and identification number (ID samples) is shown in Table 3.

### 3.2. Antimicrobial Resistance Profiles of the Isolated Bacteria

Antimicrobial susceptibility testing was conducted on 45 colistin-resistant strains, and the results are reported in Figure 1. The 32 tested antibiotics are grouped into 12 classes (Figure 1). Thirty-three (33/45; 73%) strains showed a multidrug resistance (MDR) (Figure 1). Among these, 77% (20/26) were isolated from raw food and 68% (13/19) from RTE. No statistically significant difference was observed between the two groups (*p* > 0.05). Multidrug resistance incidence was 3.3% (33/1000) in the analyzed food samples.

Notably, four strains were resistant to more than 11 antibiotics and more than four antimicrobial classes. Among these strains, S121 and S075, isolated from raw food samples, showed phenotypic resistance to nine and seven antimicrobial classes, respectively, and to 18 and 12 types of antibiotics. Their resistance patterns were as follows: AMP, AUG2, A/S2, GEN, STR, TOB, CHL, FAZ, FOX, AXO, MIN, TET, FIS, SXT, AZT, FOT, TAZ, COL and AMP, PIP, GEN, STR, TOB, CHL, NAL, MIN, TET, FIS, SXT, and COL, respectively. S123 and S084, isolated from the RTE food samples, exhibited phenotypic resistance to five antimicrobial classes and to 12 and 13 types of antibiotics, respectively. Their resistance patterns were as follows: AMP, PIP, AUG2, A/S2, TIM2, FAZ, FOX, AXO, AZT, FOT, TAZ, COL and AMP, PIP, AUG2, A/S2, P/T4, TIM2, FAZ, FOX, AXO, AZT, FOT, TAZ, and COL, respectively.

A decrease of two or three dilutions in the minimum inhibitory concentration (MIC) of the cephalosporins cefotaxime and/or ceftazidime in combination with clavulanic acid, compared to the cephalosporin alone, indicated an ESBL-producing strain. Two MDR isolates (S039 and S121) were identified as ESBL producers. All *Enterobacter* spp. and *Citrobacter freundii* isolates showed intrinsic resistance to AMP, AUG2, A/S2, FAZ, and FOX.

The highest resistance rate was observed for ampicillin, which reached 84% (*n* = 38), followed by resistance to cefazolin (*n* = 33; 73%), amoxicillin/clavulanic acid, ampicillin/sulbactam and cefoxitin (*n* = 28; 62%), and sulfisoxazole (*n*= 7; 16%). For the other sixteen antimicrobials, the resistance rates were below 10% (Figure 2). No resistance was detected to carbapenems, cefepime, piperacillin/tazobactam, ciprofloxacin, levofloxacin, and tigecycline.

### 3.3. Detection of AMR, Virulence, and Plasmid Genes

In this study, several antimicrobial resistance genes (ARGs) were identified in the isolates, using the CARD database included in ABRIcate tool. A total of 56 acquired known ARGs were categorized into nine classes based on the type of antibiotic resistance (Figure 3). Among the identified classes, β-lactams were the most abundantly found in 36 strains (36/45; 80%). Interestingly, the gene (*bla_TEM-1_*) associated with broad-spectrum beta-lactamase (BSBL) and two genes (*bla_SHV-12_* and *bla_SHV-145_*) associated with extended-spectrum beta-lactamase (ESBL) were found in three strains (S075, S039, and S121). Three genes (*fosA2*, *fosA5*, and *fosA6*), conferring resistance to Fosfomycin, were carried by 31 isolates (31/45; 69%). Notably, seven isolates harbored plasmid-mediated genes *mcr-1*, *mcr*-*9*, and *mcr*-*10*, conferring resistance to colistin. Additionally, ARGs associated with aminoglycosides, phenicols, macrolides, fluoroquinolones, sulfonamides, and tetracyclines were identified (Figure 3). A concordance rate of 100% between the genotype and phenotype (both genotype and phenotype were positive) was detected for β-lactamase (38/38), sulfonamides (10/10), and tetracyclines (5/5) (Figure 3).

Other elements, not reported in Figure 3, contributing to colistin resistance were predicted in 31/45 strains, including two chromosomally encoded intrinsic resistance genes for colistin (*eptB* and *ArnT*), as well as *kpnE* and *kpnF*, which confer resistance not only to colistin but also to macrolides, aminoglycosides, cephalosporins, tetracycline, and rifamycins through an antibiotic efflux mechanism. This study also predicted 72 elements as MDR transporters, antibiotic efflux pumps, and transcriptional activators of genes involved in the chromosomal multiple antibiotic resistance phenotype, conferring multidrug resistance and resistance to disinfectants and antiseptics (Appendix A).

Plasmids play a key role in the spread of ARGs both within and between bacterial taxa [33]. Point mutations associated with antimicrobial resistance were not identified by resistome prediction performed using the CARD database. The prediction of plasmid types among the strains was performed using plasmidFinder, and the results are reported in Appendix A. A total of 31 replicons, or part of them, were detected, and among these, ColRNAI_1 was the most dominant plasmid, identified in 18 strains, followed by Col440II_1 and Col440I_1, identified in 14 and 11 strains, respectively. Additionally, two strains (S072 and S120) exhibited the highest number of replicon types (*n* = 8), whereas twelve strains (S059, S071, S073, S076, S079, S080, S081, S093, S094, S114, S132, and S140) did not harbor any replicons. However, it should be specified that, in the strains analyzed, it was not possible to identify the entire nucleotide sequence of the plasmid but only parts of it that the database noted as being traceable to a plasmid.

The prediction of virulence genes was performed using the Virulence Factors Database (VFDB), and the results are shown in Appendix A. In total, eighty-seven virulence genes implicated in several mechanisms of virulence and pathogenicity were predicted. Independently of the strain species, the most frequent gene found (84%) was *omp*A, which encodes for an outer membrane protein. The number of virulence genes per strain is shown in Figure 4.

## 4. Discussion

The Gram-negative bacteria belonging to the *Enterobacteriaceae* family isolated in this study are part of the normal flora of the intestinal tract or environment of animals; however, these bacteria have been associated with opportunistic infections in humans [34]. In corroboration of this assumption, a recent study [35] showed that, in addition to pathogenic AMR bacteria, commensal bacteria may also pose a foodborne hazard. These bacteria, which may possess AMR genes, are widespread and, consequently, may contaminate ready-to-eat food in greater numbers than pathogenic bacteria. Such bacteria can persist in the human microbiome [36] and pose a food safety hazard [37], particularly for vulnerable individuals. In this study, among 1000 raw and RTE food samples examined, we identified 44 samples positive for colistin-resistant *Enterobacteriaceae*, notifying a prevalence of 4.4%. Our finding was in line with a previous report [38] that found a prevalence of 4.79% of colistin-resistant *E. coli.* Particularly, the highest prevalence of colistin resistance was found in Latin America (3.24%), Asia (8.67%), and Africa (5.04%); a lower prevalence was observed in Europe (1.87%) and North America (2.05%). Among the 44 food samples, a total of 45 strains were isolated (26 from raw food sources and 19 from RTE sources). Bacteria isolation and genomic sequencing identified eight bacteria genera, and the most commonly isolated genus was *Enterobacter* spp. with a rate of 60% (27/45) of the isolates, accordingly with a previous report [39]. Colistin-resistant *Enterobacter* spp. have emerged in the last decade, and based on a recent study, the British Society for Antimicrobial Chemotherapy (BSAC) revealed that the colistin resistance among *Enterobacter cloacae* complex isolates from 2011 to 2017 was 4.4–20%, which was higher than that observed in other genera, such as *Klebsiella* spp. and *Escherichia coli* [40]. With a lower prevalence, *Moellerella wisconsensis* (15.5%); *Atlantibacter hermannii* (8.8%); *Klebsiella* spp. (*K. variicola* and *K. pneumoniae*) (4.4%); and *E. coli*, *C. freundii*, *R. ornithinolytica*, *P. diazotrophicus*, and *Kosakonia cowanii* were identified in 2.2% of strains, respectively. *Moellerella wisconsensis* and *Klebsiella variicola* rarely cause human [41,42] and animal infections [42]; interestingly, colistin resistance in *Moellerella wisconsensis* has been reported in the literature, and this should be considered in daily clinical practice [41,43]. Due the widespread occurrence of colistin-resistant *K. pneumoniae* [44], in contrast to what has been described for *K. variicola* strains [45], our results suggest that the potential acquisition of colistin resistance should be monitored, particularly since colistin is the treatment of choice for carbapenem-resistant Klebsiella infections. This aspect is further stressed if we consider that we also isolated strains of *C. freundii* and *Atlantibacter hermannii* that were found resistant to colistin [46,47,48].

Additionally, we isolated a MDR *E. coli* strain from turkey meat. *E. coli* is intrinsically susceptible to antibiotics; however, it can acquire resistance determinants associated with motile genetic elements as extended-spectrum β-lactamases genes, aminoglycosides, quinolones resistance genes, and *mcr* genes [49]. The *mcr*-1-positive *E. coli* strains have been previously associated with poultry [50,51]. Account should also be taken of some strains, *K. cowanii*, *P. diazotrophicus*, and *R. ornithinolytica*, found with a low prevalence that, although they rarely cause infections in humans, raises the question of the emergence of antimicrobial resistance that could make them potential multidrug-resistant pathogens, as already observed for *Phytobacter* strains [52,53].

All the strains isolated in our study were also screened for antimicrobial susceptibility to 32 antibiotics, categorized into 12 classes. The highest resistance levels were observed with β-lactam antibiotics: ampicillin (84%), cefazolin (73%), amoxicillin–clavulanic acid (62%), ampicillin/sulbactam (62%), and cefoxitin (62%). Comparable findings have been reported in other studies, where *Enterobacteriaceae* exhibited resistance rates to ampicillin (66%), cephalothin (57% or 79.7%), amoxicillin–clavulanic acid (33% or 33.3%), and cefoxitin (31% or 36.1%) [38,54]. In our study, multidrug resistance (MDR) incidence was 3.3% in raw and cooked food samples. Previous research noted a MDR incidence of 1.9% in raw and cooked food samples in China [55], 0.6% of fresh vegetables in Spain [56], 16% of fresh fruits and vegetables in Arabia [57], and 50% of ready-to-eat foods from community canteens in Rome, Italy [38]. Additionally, genomic analysis revealed 56 antimicrobial resistance genes (ARGs) that could be consistent with the AMR phenotype. The ARGs were grouped into nine classes, based on antibiotic resistance. The most frequently detected class was β-lactams, followed by quinolones, Fosfomycin, aminoglycosides, phenicols, polymyxins (*mcr*), sulfonamides, tetracyclines, and macrolides. Additionally, 72 elements were detected as MDR transporters, antibiotic efflux pumps, and transcriptional activators of genes involved in the chromosomal multiple antibiotic resistance phenotype. Notably with regards to colistin resistance, seven isolates harbored the mobile colistin resistance (*mcr*) gene. One *E. coli* isolate carried the *mcr-1.1* gene, while six *Enterobacter* spp. strains carried the *mcr*-*9.1* and *mcr*-*10.1* genes; these genes are frequently found in *Enterobacter* spp. [57]. Particularly, the *E. coli* from a turkey meat sample and *E. kobei* from fresh egg pasta coharbored beta-lactamase-producing genes and ARGs related to aminoglycosides, tetracyclines, phenicols, sulfonamides, and trimethoprim. The colistin resistance associated with the acquisition of genes *mcr*, carried by plasmids, also played an important role in colistin resistance and represents a potential danger to human health [12,14]. Livestock, poultry, and related food products have been identified as a reservoir for the spread of *mcr* genes carrying *Enterobacteriaceae* into humans [15]. According to EFSA (2020), colistin resistance is notably more prevalent in MDR *E. coli* from poultry (2.7%) and turkeys (9.5%) compared to pigs (0.2%) and calves (0.4%) [11]. In the literature, in addition to the *mcr* gene, other various mechanisms of colistin resistance in *Enterobacteriaceae* have been described (lipopolysaccharide modifications, efflux pumps, capsule formation, and overexpression of membrane proteins) [58]. In our study, 31 out of 45 isolates harbored the *eptB*, *ArnT*, and two other MDR genes (*KpnE* and *KpnF*) associated with colistin resistance.

When we compared phenotypically antimicrobial resistance with predictions based on genomic sequencing, we noticed some concordance. All *Enterobacter* spp. strains exhibited phenotypic resistance to first-generation cephalosporins and cefoxitin, which can be explained by the prediction of AmpC beta-lactamases (*bla_ACT_* and *bla_CMH_*) genes. All *A. hermannii* strains exhibited phenotypic resistance to ampicillin, and in each of them was predicted the *bla_HERA_* gene, indicative of resistance to ampicillin. The *C. freundii* strain showed resistance to ampicillin, first-generation cephalosporins, and cefoxitin, and the same strain harbored the AmpC (*bla_CMY_*) gene, which was related to this genus [59]. Both *K. variicola* and *R. ornithinolytica*, which harbored the *bla_LEN-12_* and *bla_ORN-1_* genes, respectively, were phenotypically resistant to ampicillin. The *E. coli* strain that showed resistance to ampicillin and first-generation cephalosporins harbored the broad-spectrum β-lactamases gene *bla_TEM-1_*. Finally, *K. pneumoniae* and *E. kobei* (S121) harbored the ESBL genes (*bla_SHV-145_* and *bla_SHV-12_*) in accordance with the phenotypes exhibited. Further accordance, although partially, was also found for other antibiotic resistance. Aminoglycoside resistance genes (*aac, aph, aad,* and *ant*) were found in six isolates (13%), including *E. coli*, *E. kobei*, and *M. wisconsensis*, of which three showed phenotypic resistance to streptomycin, tobramycin, and gentamicin. Chloramphenicol resistance genes (*catA* and *clmA*) were detected in six isolates (13%), including *E. coli*, *E. kobei*, and *A. hermannii*; among these, two strains were also phenotypically resistant to chloramphenicol. Sulfonamide resistance genes (*sul*) were found in five isolates (11%), all phenotypically resistant to sulfisoxazole. An exception is the *C. freundii* strain that showed phenotypical resistance to sulfisoxazole but did not carry any gene associated with this resistance. Genes associated with tetracycline (*tet*) and trimethoprim (*dfrA*) resistance were predicted in four (9%) and three strains (7%), respectively, in accordance with phenotypic resistance. Conversely, antimicrobial resistance genes (i.e., *qnr* gene associated with quinolone resistance and *ereA* gene associated with macrolide resistance) have been identified in some strains that were phenotypically susceptible. Unfortunately, the Fosfomycin resistance gene (*fosA*) was found in twenty-nine (64%) isolates, but we did not correlate this result with phenotypical resistance, since susceptibility to the corresponding antibiotic was not tested. No AMR genes were found in eight isolates (18%), including *Phytobacter diazotrophicus*, *Kosakonia cowanii*, and *M. wisconsensis*. It is evident that a significant correlation between the presence of ARGs and phenotypic resistance was observed particularly for β-lactamases, sulfonamides, and tetracyclines. However, the acquired ARGs in bacterial genomes do not necessarily confer phenotypic resistance and vice versa [60,61]. Other mechanisms not examined in this study, such as antimicrobial resistance conferred by single-nucleotide polymorphisms (SNPs), can contribute to the phenotypic resistance [60].

Regarding the pathogenic role exercised by the isolated strains, genomic sequencing revealed a total of 87 virulence genes (VGs) associated with various mechanisms of virulence and pathogenicity. In particular, the *E. coli* strain carried the highest number of virulence genes (84 VGs), indicating a potentially high pathogenicity. Two *E. cancerogenus* strains carried 14 VGs, and the *K. pneumoniae* harbored 10 VGs. The remaining strains harbored fewer VGs, ranging from 1 to 9. In concordance with another study [62], all strains, with the exception of all *M. wisconsensis*, carried the *ompA* gene, which encodes for an outer membrane protein that mediates bacterial biofilm formation, eukaryotic cell infection, antibiotic resistance, and immunomodulation [62]. Although we have not carried out functional studies to assess the virulence of the isolated strains, the data obtained, corroborated from data available in the literature, suggest that, with exceptions of *M. wisconsensis*, the strains described may be potentially pathogenic.

In this study, plasmidFinder identified a total of 31 replicons, or parts of replicons, in the strains analyzed. Twelve strains did not harbor any identifiable plasmids. Among the 45 strains, 20 harbored Inc plasmids, capable of carrying several AMR genes, including those associated with ESBLs and *mcr* genes [18,63]. Particularly, in four of the seven isolates harboring the *mcr* gene, we found parts attributable to the IncHI2 plasmids. These plasmids are known for their role in the global dissemination of the *mcr* gene in *Enterobacteriaceae*.

## 5. Conclusions

This study contributes data on the prevalence of multidrug-resistant *Enterobacteriaceae* in various food sources, including RTE foods. It shows the presence of colistin resistance and ARGs in strains, some of which are known to cause foodborne infections; particularly, the results pose reflections on the role of commensal bacteria as potential reservoirs for AMR genes. It highlights the presence of multiple resistance mechanisms, including the *mcr* genes, in various bacterial species and underscores the potential risk posed by these bacteria to human health. The presence of the *mcr*-*1* gene in *E. coli* and *mcr-9* and *mcr-10* genes in *Enterobacter* species suggests a horizontal transfer of resistance, often facilitated by plasmids. Our results focus attention on two other genes, *eptB* and *ArnT,* that may play a role in colistin resistance, but further studies are needed to understand their molecular mechanism. Additionally, it is probable that other unknown genes encode proteins or enzymes associated with colistin resistance. It is known, for example, that ARGs map on plasmid sequences; particularly, IncHI2 cause significant concern, because they are known to transport and facilitate the global spread of *mcr* genes in *Enterobacteriaceae*. Our study revealed that four of the seven strains harboring the *mcr* gene contained parts of IncHI2 plasmids, but the genome sequencing performed was not sufficient to completely characterize the plasmids. Consequently, further efforts are needed not only to improve the knowledge of the association between specific genes and colistin resistance but also more comprehensive genomic data, enabling the assembly of complete plasmids. Thus, the application of molecular assays for the rapid and accurate detection of resistance determinants is crucial for reducing and controlling MDR bacterial infections and for understanding the epidemiology and dynamics of spreading AMR genes. Combining genomic analysis with phenotypic testing provides a more comprehensive understanding of bacterial resistance mechanisms, which is essential for informing effective treatment strategies. To counter the antibiotic resistance spread is absolutely necessary to allocate more resources to formulate more effective prevention and contrast plans.

## Figures and Tables

**Figure 1 microorganisms-13-00163-f001:**
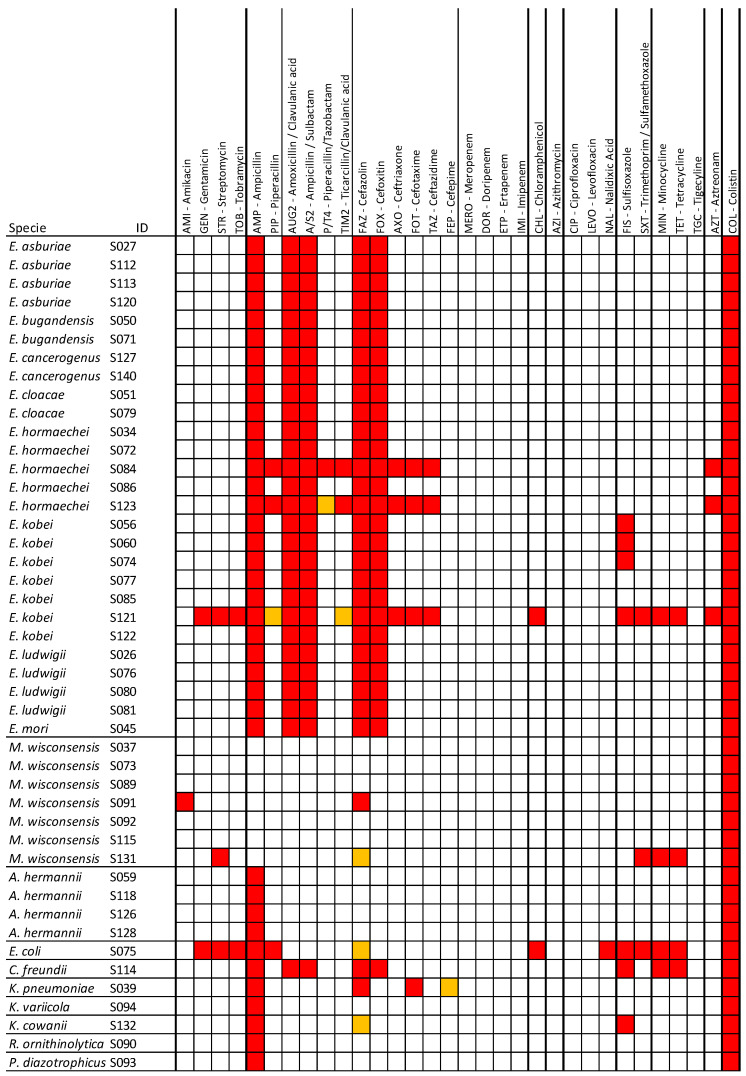
Schematic view of the antimicrobial resistance results of 45 strains of *Enterobacteriaceae*, and which species is indicated in the first column. Antimicrobial agents, grouped by drug class, detected in the MIC are listed in the columns. Resistance is indicated in red, susceptibility in white, and intermediate resistance in orange.

**Figure 2 microorganisms-13-00163-f002:**
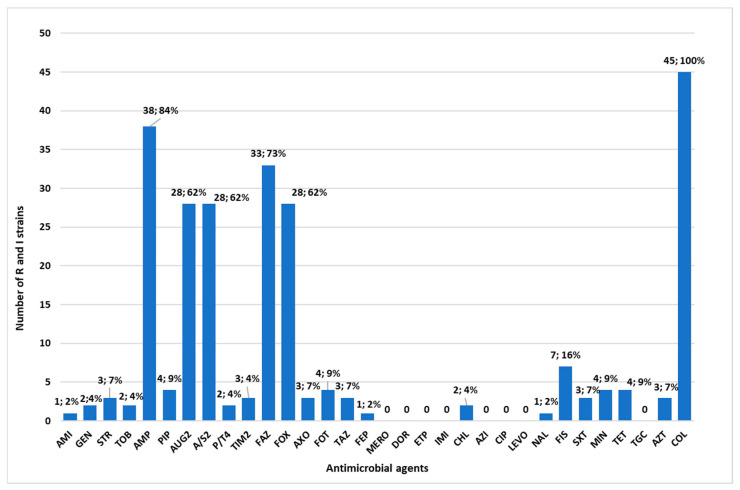
Number and percentage (*n*; %) of resistant (R) or intermediate-resistant (I) strains to the tested antibiotics.

**Figure 3 microorganisms-13-00163-f003:**
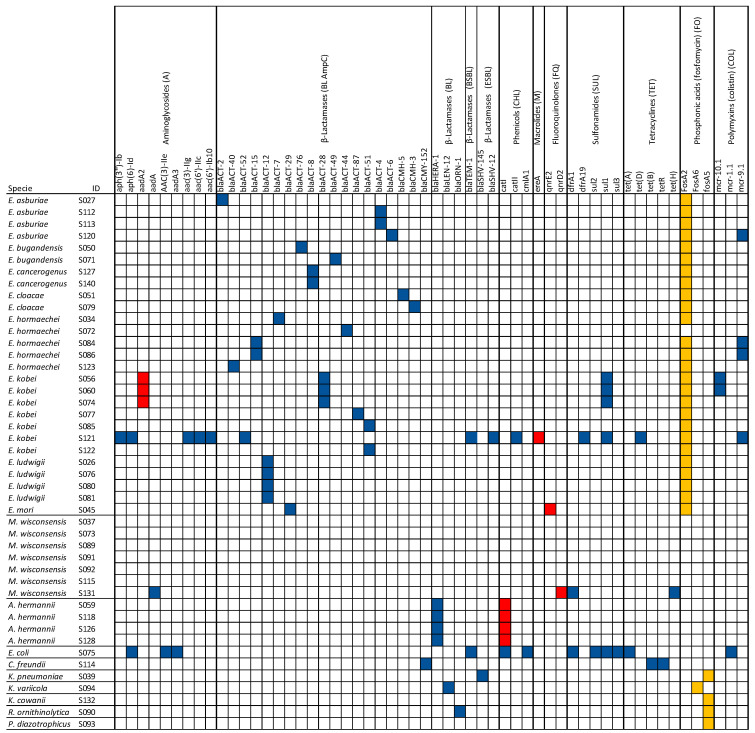
Schematic view of resistomes related to the 45 strains (S#ID). The first column lists the species predicted by WGS. Additionally, the table reports the AMR genes identified by both WGS and in vitro analyses. Strains that harbored AMR genes, identified by WGS and correlated with a specific resistance confirmed by in vitro analysis, are highlighted in blue. AMR genes correlated with specific resistance that were not confirmed by in vitro analysis are indicated in red. Predicted genes not detected by in vitro analysis are shown in orange.

**Figure 4 microorganisms-13-00163-f004:**
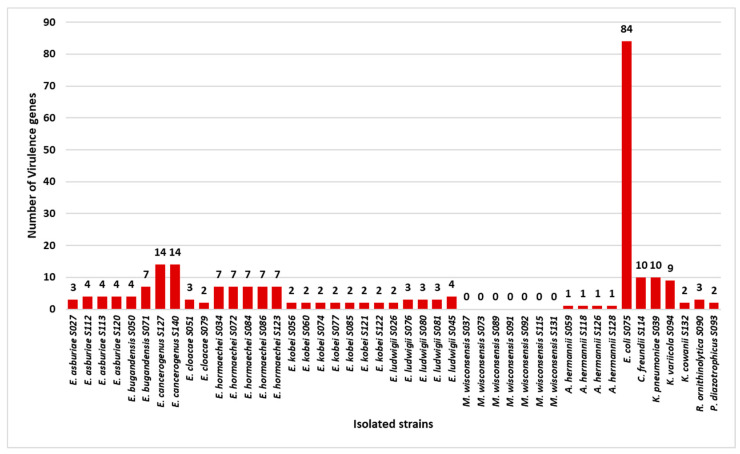
Number of virulence genes per strain (grouped by species).

**Table 1 microorganisms-13-00163-t001:** Colistin-resistant *Enterobacteriaceae* isolated from raw food samples. N. = number. ** E. kobei* and *E. cancerogenus* were isolated from the same fresh egg pasta sample.

Source	N. of Samples	Number of Positive Food Samples	Percentage of Positive Samples for Type of Food (%)	N. of Isolates	Organism (*n*., Name of Food)
Raw Milk	52	2	4	2	*E. hormaechei* (1, bulk cow milk)
					*E. mori* (1, bulk cow milk)
Raw Meat	207	11	5	11	*E. ludwigii* (1, bovine meat) *E. bugandensis* (1, minced beef/pork)
					*E. kobei* (1, fresh pork sausage, 1, sausage dough)
					*E. asburiae*(1, pork meat)
					*M. wisconsensis* (1, beef meat; 1, minced beef; 1, pork fresh sausages; 1, beef and pork hamburgher; 1, poultry meat)
					*E. coli* (1, turkey meat)
Raw Seafood products	133	4	3	4	*A. hermannii* (1, filleted sea bass)
					*E. hormaechei* (1, sea bass)
					*M. wisconsensis* (1, filleted sea bass)
					*R. ornithinolytica* (1, clams)
Raw Bakery and pastry products, fresh pasta	28	2	7	3	*E. cancerogenus* (1, raw shortcrust pastry; 1, fresh egg pasta *)
					*E. kobei* (1, fresh egg pasta *)
Vegetables	80	6	8	6	*E. bugandensis* (1, beets)
					*E. cloacae* (2, cucumbers)
					*E. ludwigii* (1, celery)
					*E. kobei* (1, frozen celery)
					*K. cowanii* (1, wheat flour)
TOT	500	25	5	26	

**Table 2 microorganisms-13-00163-t002:** Colistin-resistant *Enterobacteriaceae* isolated from RTE food samples N. = number.

Source	N. of Samples	Number of Positive Food Samples	Percentage of Positive Samples for Type of Food (%)	N. of Isolates	Organism (*n*., Name of Food)
Milk and Cheese	240	8	3	8	*E. asburiae* (1, burrata cheese)
					*E. kobei* (1, mozzarella; 1, sheep and goat cheese; 1, giuncata cheese)
					*E. hormaechei* (1, fresh cheese; 1, giuncata cheese)
					*A. hermannii* (1, mozzarella; 1, fresh cheese)
Dried or cooked Sausages	35	1	3	1	*A. hermannii* (1, cured sausage)
Ready meals	100	3	3	3	*E. ludwigii* (1, seasoned salad)
					*C. freundii* (1, grilled swordfish with olives)
					*M. wisconsensis* (1, breaded salmon fillet)
Bakery and pastry products, fresh pasta	27	0	0	0	0
Ice-cream	66	5	8	5	*K. pneumoniae* (1, homemade ice-cream)
					*E. asburiae* (2, homemade ice-cream)
					*E. hormaechei* (1, homemade ice-cream)
					*P. diazotrophicus* (1, homemade ice-cream)
Vegetables	32	2	6	2	*K. variicola* (1, IV range mixed salad)
					*E. ludwigii* (1, IV range fruit salad)
TOT	500	19	4	19	

**Table 3 microorganisms-13-00163-t003:** In the table are collected sources (raw food and RTE samples), organisms, grouped for species, and ID strain. * *E. cancerogenus* S140 and *E. kobei* S121 were isolated from the same sample.

**Raw Food Samples**	**Organism**	**ID Samples**
**Source**	**Species**	
Raw turkey meat	*E. coli*	S075
Raw sea bass fillet	*A. hermannii*	S128
Raw clams	*R. ornithinolytica*	S090
Raw sea bass fillet	*M. wisconsensis*	S131
Raw bovine meat	*M. wisconsensis*	S037
Raw poultry meat	*M. wisconsensis*	S073
Fresh pork sausage	*M. wisconsensis*	S091
Raw minced bee	*M. wisconsensis*	S092
Raw hamburger	*M. wisconsensis*	S089
Wheat flour	*K. cowanii*	S132
Raw pork meat	*E. asburiae*	S027
Raw minced beef and pork	*E. bugandensis*	S071
Beets (no rte)	*E. bugandensis*	S050
Cucumber (no rte)	*E. cloacae*	S051
Cucumber (no rte)	*E. cloacae*	S079
Raw shortcrust pastry	*E. cancerogenus*	S127
Fresh egg pasta *	**E. cancerogenus*	S140
Bulk cow milk	*E. hormaechei*	S034
Raw sea bass	*E. hormaechei*	S086
Raw bovine meat	*E. ludwigii*	S026
Celery (no rte)	*E. ludwigii*	S076
Bulk cow milk	*E. mori*	S045
Raw fresh pork sausage	*E. kobei*	S060
Raw sausage dough	*E. kobei*	S085
Celery (no rte)	*E. kobei*	S077
Fresh egg pasta *	** E. kobei*	S121
**Rte Samples**	**Organism**	**ID Samples**
**Source**	**Species**	
Homemade ice cream	*K. pneumoniae*	S039
Mixed salad (rte)	*K. variicola*	S094
Mozzarella	*A. hermannii*	S118
Fresh cheese	*A. hermannii*	S126
Seasoned sausage (rte)	*A. hermannii*	S059
Grilled swordfish (rte)	*C. freundii*	S114
Breaded salmon fillet (rte)	*M. wisconsensis*	S115
Homemade ice cream	*P. diazotrophicus*	S093
Burrata	*E. asburiae*	S120
Homemade ice cream	*E. asburiae*	S112
Homemade ice cream	*E. asburiae*	S113
Fesh cheese	*E. hormaechei*	S072
Seasoned salad (rte)	*E. ludwigii*	S081
Fruit salad (rte)	*E. ludwigii*	S080
Mozzarella	*E. kobei*	S056
Sheep and goat cheese	*E. kobei*	S074
Giuncata	*E. kobei*	S122
Giuncata	*E. hormaechei*	S123
Homemade ice-cream	*E. hormaechei*	S084

## Data Availability

The data that support the findings of this study are available from the corresponding author upon reasonable request.

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
