# Peer review of "Isolation and Characterization of Colistin-Resistant Enterobacteriaceae from Foods in Two Italian Regions in the South of Italy"

_microorganisms, 2025, doi:10.3390/microorganisms13010163_

Round 1

Reviewer 1 Report

Comments and Suggestions for Authors

The manuscript “Isolation and characterization of colistin-resistant Enterobacteriaceae from foods in two Italian regions in the South of Italy” presents interesting information. In the course of reviewing the presented results, some questions and comments arose, listed below.

Comments:

Lines 48-51. References should be in the end of the sentence.

Line 150. “45 colistin-resistance” should be replaced on “45 colistin-resistant”.

Tables 1-3. Bacterial names should be presented as “E. ludwigii”, “E. asburiae”, etc. after the first use.

Line 190 and others. “spp.” should be presented in non-italic.

Line 211. Beta-lactamase genes should be presented as “blaSHV-1”. SHV-1 and TEM-1 belong to the broad-spectrum class A beta-lactamases. Only SHV-12 is the extended-spectrum class A beta-lactamase. Why these genes are not presented in Table 6?

Tables 4, 6. Why are the isolates not grouped by species? Why are all the isolates named ISBL, although most of them do not have a phenotype that corresponds to ISBL?

I propose to rename these Tables as Figures 1, 3.

Figure 2. Why are the isolates not grouped by species?

Discussion. There is no comparison of isolates obtained in two regions of Italy, although this is included in the title of the manuscript.

Lines 404-410. It is not clear which groups of pathogenicity genes have been identified in which species. Can these bacteria be considered potentially dangerous?

Conclusions. What is the novelty of this study? Has the underlying mechanism of colistin resistance been identified? What is the potential of the plasmids identified?

Reviewer 2 Report

Comments and Suggestions for Authors

- Abstract should rewrite, Like 1.000-->1,000 (line 16), then check whole manuscript about the number or samples and another number either

- Keywords should arrange in alphabetically

- In Introduction: The objectives should rewrite to attract reader more

- Figure number and legends should confirm in between 225-226

- Discussion part should rewrite to make it more feasible and readble. Too many paragraphs.

- Gene in written format follow the same format "italic"

- Overall English should improve

- High Plagiarism (27%), should less than 15-20%.

Comments on the Quality of English Language

- Overall English should improve
